# Looking Inside the World of Granulosa Cells: The Noxious Effects of Cigarette Smoke

**DOI:** 10.3390/biomedicines8090309

**Published:** 2020-08-27

**Authors:** Fani Konstantinidou, Liborio Stuppia, Valentina Gatta

**Affiliations:** 1Department of Psychological, Health and Territorial Sciences, School of Medicine and Health Sciences, “G. d’Annunzio” University of Chieti-Pescara, 66100 Chieti, Italy; fanikonst@hotmail.com (F.K.); stuppia@unich.it (L.S.); 2Center for Advanced Studies and Technology (CAST), “G. d’Annunzio” University of Chieti-Pescara, 66100 Chieti, Italy

**Keywords:** cigarette smoke, granulosa cells, electronic cigarettes, female reproduction, pregnancy, infertility, IVF

## Abstract

The detrimental implications of tobacco smoke on systemic health have been widely established during the past few decades. Nonetheless, increasing evidence has begun to shed more light on the serious impact that smoke exposure could also have on mammal reproductive health in terms of overall ovarian dysfunction and gestation. A variety of these complications seem to be causally related to specific chemical substances contained in cigarette smoke and their possible effects on ovarian tissues and cells, such as granulosa cells. Granulosa cells represent the functional unit of the ovary and are able to establish a bidirectional cross-talk relationship with the oocyte during folliculogenesis, which makes them vital for its correct growth and development. Based on these premises, the current review focuses on the presence of related smoke-induced damages in granulosa cells. Data have been grouped according to the studied tobacco constituents and the molecular pathways involved, in order to synthesize their impact on granulosa cells and fertility. Attention is further brought to the correlation between electronic cigarettes and female reproduction, although there have been no investigations so far regarding e-cigarette-related granulosa cell exposure. We summarize how tobacco constituents are able to cause alterations in the “life” of granulosa cells, ranging from luteal steroidogenesis and follicular loss to granulosa cell apoptosis and activation of the autophagic machinery. Further studies have been conducted to elucidate the relationship between lifestyle and fertility as to reduce the morbidity connected with infertility.

## 1. Introduction

Cigarette smoke remains to this day a global health issue, being known to increase the incidence of a series of highly impacting diseases, such as coronary heart disease, stroke, and lung cancer [1] for both male and female populations all around the world. Increasing evidence demonstrates that tobacco smoke constituents have also potential toxic effects on human and wildlife reproductive health [2,3], damaging the reproductive system with particular reference to the ovary [4]. The ovary is of vital importance in many aspects of female reproduction, such as the synthesis of steroid hormones, which are fundamental to subsequent follicular growth and oocyte maturation [5].

In the male, on the other hand, it has been reported a dose-dependent correlation between smoking, semen quality, and sperm function [6].

To better underline the mechanisms of smoke-related ovarian toxicity, the adverse consequences of cigarette smoke have furtherly been examined, both in humans and in animal models such as dairy cattle, mice, and pigs.

The main reason behind these noxious effects lies in the harmful chemical components contained in cigarettes, such as cadmium, a known heavy metal, alkaloids, widely represented by nicotine and cotinine, and benzoapyrene, a polycyclic aromatic hydrocarbon, considered a major carcinogen of cigarette smoke. Even if cadmium is considered to be a major environmental pollutant and is often found in high concentrations in plastic materials, the soil and, as a direct consequence, in animals and different types of food, such as meat and fruit, its most common way of exposure is still believed to be generated through tobacco smoke contamination [7].

Furthermore, exposure to alkaloids can have severe toxic repercussions on reproductive health by negatively impacting steroidogenesis in both humans and other large mammals, or by contributing to the destruction of the follicle population. Tobacco-containing nicotine, for example, is an easily and rapidly absorbed substance by the organism through the respiratory system [8], making it one of its major active components. For this reason, it could potentially have a lot of systemic side effects concerning vital organs, like the heart, lungs, and kidneys, but also the reproductive tissues and cells.

Lastly, it has been shown that benzoapyrene, a known carcinogenic constituent of cigarette smoke with considerable mutagenic properties, can potentially cause a primordial follicle reduction, follicular atresia, and overall ovotoxicity in mammals.

On the whole, metals such as lead, mercury, nickel carbonyl, and inorganic oxides of arsenic have all been listed as known factors of reproductive toxicity, possibly contributing to adverse effects of tobacco smoking on reproduction [9]. Studies have indicated associations between prenatal exposure to lead and spontaneous abortion, preterm delivery, and reduced birth weight [10], as well as to fetotoxins present in tobacco smoke, like PAHs, Aryl Hydrocarbon (AH), and benzene, equally contributing to adverse pregnancy outcomes by inducibility of phase I enzymes, such as CYP1A1 [11].

Nonetheless, in the current review attention will be primarily payed to heavy metal cadmium, alkaloids such as nicotine, cotinine and anabasine, and benzoapyrene, considering existing literature following our targeted search, which specifically indicates a significant correlation between these smoke components and mammal granulosa or cumulus cells.

These constituents have been widely explored for their influence on reproduction and fertility complications. Several studies have also focused their attention on ovarian granulosa cells (GCs).

GCs, the functional unit of the ovary, play a pivotal role in the regulation of ovarian reproductive health in females. They are somatic cells of ovarian origin that surround the oocyte and undergo proliferation during folliculogenesis and oocyte growth, which are drastically influenced by the underlying germ cells through a bidirectional cross-talk with the oocyte. The relationship in question is a dynamic one, especially considering that the number of granulosa cells, for instance, is well-associated with normal oocyte development [12].

They are considered to be the main target by the chemicals at all stages of their development and it has been reported that tobacco smoke constituents induce toxicity of GCs by interfering with the cell cycle. Damage to follicles due to tobacco smoke-related exposure takes place during the early stages of growth and for all the above, the importance of further examination regarding the nature of smoke’s effects on this type of cells is undoubtedly comprehensible.

In the current literature revision paper, a computer-assisted search of the PubMed and Google Scholar databases was adapted, in order to better trace all relative publications coherent to the specific subject. The following key phrases were used alone or in combination with each other: “Cigarette smoke”, “granulosa cells”, “electronic cigarettes”, “female reproduction”, “pregnancy”, “infertility”, “IVF”. This search was conducted in accordance with the Preferred Reporting Items for Systematic Reviews and Meta-Analyses (PRISMA) guidelines. Only publications written in English were considered. The initial query was also limited to articles published from year 1999 to 2020, which specifically examined the relationship between tobacco smoke or tobacco smoke constituents and mammal granulosa cells, in terms of folliculogenesis and follicular growth, steroidogenesis, granulosa cell apoptosis, and autophagy-mediated pathways, as well as the overall influence of electronic cigarettes on female reproductive health. The eligibility of the studies was firstly based on the titles and corresponding included abstracts. With the exception of articles concerning e-cigarettes, we excluded those which did not specifically concern “smoke” and “granulosa” or “cumulus cells”. Studies not related to mammals were also finally discarded (Figure 1). Considering the above, the initial percentage of titles in agreement with the search keywords has been calculated roundly at 85%. Full manuscripts were then retrieved for all selected papers and final inclusion was made after thorough examination. Two of them were deemed irrelevant, not being able to further add to the general structure of the paper. Their reference lists were also assessed, in order to detect other potentially related studies or information that could be further included in this review. Lastly, data were finally regrouped according to the different studied smoke-contained substances, molecular pathways involved, and their individual effects on mammal granulosa cells.

The main objective of this review would be to regroup and organize already existing scientific information on how cigarette smoke is able to exert a series of detrimental effects on granulosa cells in mammals, a fundamental, to oocyte development, type of reproductive cell. This kind of impact could be analyzed through cellular exposure to tobacco-contained components, such as cadmium, single alkaloids or mixtures of them, and representatives of the PAH family, or even in terms of granulosa cell apoptosis and activation of autophagy-mediated pathways.

## 2. Cigarette Smoke Chemical Components and Their Effects on Granulosa Cells

### 2.1. Cadmium Exposure

Cadmium is a chemical element that is largely diffused in the environment and into living organisms as a result of pollution originated by multiple factors. Its main source of contamination may be represented by tobacco smoke, a type of exposure steadily on the rise, especially between women of reproductive age [13] in terms of certified complications associated with human fertility. The scale of harm potentially caused by the heavy metal could be related to the fact that it possesses a quite long biological half-life, estimated approximately between 15 and 30 years, as well as its slow expulsion from the body [7], causing it to cumulatively concentrate in a time-dependent manner in various reproductive organs, such as the ovaries and placenta [14,15]. This gradual accumulation could be translated into an increased exposure risk of these organs to toxic levels of the smoke-generated cadmium, causing deleterious effects on the reproductive structures.

More specifically, it has been seen that in culture-contained granulosa cells originated from both humans and rats, this heavy metal was able to inhibit the synthesis of a steroid hormone vital to follicular growth, such as progesterone [16,17].

In reality, both stimulatory and inhibitory effects of cadmium on the synthesis of ovarian progesterone were studied using stable porcine granulosa cells. In this way, it was demonstrated that this type of exposure was able to significantly modify the transcription and overall activity of the *CYP11A1* gene, also known as P450 side-chain cleavage gene [7]. This gene encodes a member of the cytochrome P450 superfamily of enzymes that localizes to the mitochondrial inner membrane. Its protein catalyzes the conversion of cholesterol to pregnenolone, the immediate precursor of progesterone [18], providing a vital steroidogenesis-based reaction in all mammalian tissues responsible for steroid hormonal production. It has been reported that cultures of porcine granulosa cells exposed to both low (0.6–3) and high (5 μM) concentrations of CdCl_2_ react to stress in two opposite ways regarding the *CYP11A1* promoter activity. At low concentrations, CdCl_2_ exposure stimulated the transcription of the *CYP11A1* gene in a dose- and time-dependent manner, favoring the ovarian steroidogenic pathway, while at high concentrations, it inhibited gene expression and, as a consequence, the progesterone synthesis in the ovary, leading to possible modifications in cell morphology and cell death. These findings highlighted the fact that, depending on its concentration, cadmium could exert dual effects on steroidogenesis.

Nonetheless, it was observed that cigarette smoke components, such as cadmium, can also have adverse effects on the oocyte-cumulus complex (OCC) expansion [19]. Various studies attempting to establish indirect, non-invasive methods to evaluate oocyte quality [20,21] have turned their attention to cumulus cells since they are believed to mirror oocyte characteristics [22]. Among all cell types differentiating within the ovarian follicle following gonadotropin stimulation, CCs are of most importance for the development of a competent gamete [23]. They are defined as a group of closely associated granulosa cells that versus the final stages of folliculogenesis surround the oocyte and participate in the processes of oocyte maturation and fertilization. Cumulus cell function is dependent on gap junctions that form a compact structure between cumulus cells and oocytes, known as OCC.

Correct cumulus expansion is necessary to sufficiently enlarge the complex in question and help it detach from the follicular wall, in order to initiate ovulation [8]. An optimal expansion corresponds to consecutive normal ovulation, fertilization and development [24].

Still, suppression of the cumulus expansion stimulated by FSH was observed in the presence of different concentrations of cadmium. More precisely, at the highest metal concentration used, equivalent to 100 μM, a noteworthy inhibition of hyaluronic acid production on behalf of the cumulus cells was also evidenced. All of the above information indicated that OCC expansion could possibly represent another site of reproductive disruption due to exposure to noxious cigarette smoke components as reported in Figure 2.

### 2.2. Benzoapyrene Exposure

Benzoapyrene, also known as BaP, is a member of the polycyclic aromatic hydrocarbon family (PAH) and is generated through the incomplete combustion of carbon or, more specifically, fossil fuels and organic matter [25] at an interval of temperatures ranging between 300 and 600 °C. This chemical is an environmental pollutant and known toxicant, present in somewhat high concentrations in tobacco smoke [26]. In addition, it tends to form DNA adducts, commonly traced in the granulosa cells of female smokers, and is mainly activated through the action of P450 enzymes like CYP1B1, controlled by the AHR or aryl hydrocarbon receptor pathway.

A series of studies have demonstrated that exposure to environmental toxicants (ETs), such as BaP, can lead to a rapid depletion of the follicle population [27,28], which could cause premature ovarian failure (POF) [29] and an overall harmful effect on follicle development, establishing it as a potent ovotoxicant.

Considering, however, that the basic mechanisms behind mainstream cigarette smoke are not yet fully known, it was initially attempted to evaluate if smoke-generated BaP could potentially cause an increase in the *Bax*/*Caspase 3*-mediated apoptosis of ovarian follicles and, as a consequence, their granulosa cells, in murine models [30]. In the beginning, five C57BL/6 mice, between 6- and 8-weeks old, had to undergo nose-only cigarette smoke-exposure for an 8-week time period compared to 5 other mice exposed only to room air. At gross inspection, there was a difference in size and a non-statistically significant 20% inferior ovarian volume in the cigarette smoke-exposed ovaries compared to the ones belonging to their corresponding age-matched sham controls. More importantly, by microscopic examination, it was detected that there had been significant follicular number-related reductions in different stages of ovarian development in the mice exposed to tobacco smoke. Specifically, the reductions in question concerned the number of primordial ovarian follicles, while they did not seem to particularly affect primary, secondary, or antral follicles compared to their controls. Ovarian exposure to benzoapyrene in vitro led to an increase of the pro-survival factor *Bcl-2*, but there were no alterations regarding apoptosis whatsoever, with protein expression remaining unchanged for pro-apoptotic markers Bax and active Caspase 3.

Still, further examinations were necessary, especially in terms of the currently fleeting molecular aspect of ovotoxicity. For this reason, they were carried out by evaluating, for instance, the damaging effects of BaP on the ovarian transcriptome, as well as the ones of its in vivo exposure on oocyte dysfunction in mice [31]. The study in question took place by administering to female Swiss mice 7 consecutive doses a day of sesame oil either containing acetone (<10 μL/kg) or different doses of benzoapyrene, equivalent to 1.5 and 3 mg/kg correspondingly. Concerning the impact of this chemical on the neonatal ovarian transcriptome, microarray analysis indicated the existence of a complex BaP-causally-related ovotoxicity mechanism, involving as targets a series of genes, which were associated via ingenuity pathway analysis (IPA) software mainly to vital reproductive functions, such as follicular development and progression of the cell cycle (like *Igf2* and *Hspa8*), as well as cell cycle arrest and cell death (like, for example, *Cdkn1a*). Histomorphological and immunohistochemical analyses confirmed these findings and additionally suggested that, following exposure, there was also an increase in the primordial follicle activation and development of follicular atresia both in vivo and in vitro at an ovarian level. Finally, as the last part of these BaP-induced ovotoxicity processes, functional analysis focused on general oocyte dysfunction, detected severe mitochondrial damage through ROS production, oolemma lipid peroxidation, and a significantly reduced sperm–egg binding capacity as a result of brief exposure to benzoapyrene, highlighting how this chemical compound could partially be responsible for the cited effects of cigarette smoke on follicular growth and possible impaired fertilization ability in adulthood, as it is also summarized in Figure 2.

### 2.3. Alkaloid-Related Exposure

Alkaloids are a known class of organic compounds, mostly composed of carbon, hydrogen, and nitrogen atoms. They are naturally produced by a vast spectrum of organisms, like animals and large mammals, but also bacteria, fungi, and plants [32]. These kinds of molecules are also known to interact with cannabinoid receptors, even though the nature of their cannabinoidergic action requires further investigations. Still, benzophenanthridine alkaloids chelerythrine and sanguinarine have shown potential cannabimimetic properties through the inhibition of the CB1R activation following agonist treatment [33]. Tobacco leaves are undoubtedly characterized by a high content of alkaloid chemicals [9], with the most diffused one being nicotine, followed by some of its main predominant metabolites, nornicotine, cotinine, and anabasine. Cotinine, in fact, possesses a longer biological half-life, estimated around 16–19 h based on the body fluid of origin [34], which automatically makes it a utilizable marker for recent cigarette smoke exposure [35]. In addition, both nicotine and minor secondary amine alkaloids are well-established precursors of carcinogenic tobacco-specific nitrosamines or TSNAs [9]. Potential targets of this group of toxicants are represented by the nervous system, kidneys, liver, heart, but also reproductive organs.

For this reason, there have been multiple studies evaluating the negative impact of these cigarette smoke constituents on intrafollicular processes or, as a consequence, fecundity and female fertility [8].

Steroidogenesis is considered to be one of these vital ovarian-based processes, occurring in order to synthesize a large number of steroid hormones starting from cholesterol, such as estrogens and progesterone, necessary for follicular growth, oocyte maturation, and ovulation. Nicotine was able to affect basal luteal steroidogenesis in human luteal granulosa cells, but without having an impact on progesterone release induced on behalf of hCG. Furthermore, it was deemed capable of notably increasing the levels of the luteal cell-generated luteolytic prostaglandin or F-2α, while inhibiting, at the same time, the luteotropic PGE-2. Lastly, it was shown that this specific alkaloid could also increase the mRNA expression of the vascular endothelial growth factor, an important contributor in luteal pathophysiology. All the above suggested that alkaloid chemical substances, such as nicotine and its primary derivates, were responsible of two main findings. By modulating the prostaglandin (PG) system, they seemed to provoke a luteal insufficiency in humans as a result of having severely inhibited progesterone release. This inhibition of progesterone production due to tobacco smoke alkaloid-exerted effects, appeared to be causally-related to growth prevention and/or killing of the steroid-synthesizing cell [36].

Moreover, it was reported that incubation of human granulosa cells with a series of well-known tobacco alkaloids, such as cotinine, anabasine, a combination of these two major metabolites, and nicotine, as well as an aqueous tobacco smoke extract in their corresponding growth media for 48 h, led to an overall inhibition of progesterone synthesis, while their effects on estradiol were somewhat different [37]. Cotinine and anabasine slightly stimulated the normalized estradiol production, but nicotine, the mixture of all three alkaloids, and cigarette smoke extract had no impact on normalized estradiol synthesis at all. This could be explained by the fact that the latter are defined by a higher toxicity compared to cotinine and anabasine, which inevitably resulted in non-increasing estradiol levels.

Cells isolated from bovine ovaries have additionally been used in order to test the hypothesis of the nicotine- and cotinine-related inhibition of steroid production on behalf of the granulosa cells in vitro. More specifically, bovine granulosa cells were cultured with nicotine or cotinine for 24 h and estradiol concentrations were measured in the culture medium thanks to the use of double-antibody radioimmunoassays [38]. It was demonstrated that nicotine was able to inhibit estradiol production, but only at its highest tested dose (600 μM), decreasing the steroid hormone’s concentration to 12% of control values. This decrease was unquestionably caused due to nicotine-exerted cytotoxic effects on the granulosa cells, taking into consideration that there were no remaining viable cells at the end of the culture period. Cotinine, on the other hand, did not influence the estradiol production on behalf of the bovine granulosa cells.

As far as other important reproductive processes are concerned, it has been observed in granulosa cells isolated from porcine ovarian follicles that nicotine was able to potentially exert a significant effect on the OCC expansion alongside other harmful cigarette-contained components, by specifically contributing to the retention within the extracellular matrix of the oocyte-cumulus complex itself [14]. In this same context, it was also highlighted that when nicotine was present in low concentrations (<0.5 mM), it had no noticeable effect on the process of oocyte maturation whatsoever. When the oocytes were exposed to higher concentrations of this specific alkaloid (5 mM), nevertheless, significant perturbations at both the first and second meiotic division were evidenced, progressively resulting in atypical chromosome configurations [39].

The effect of nicotine has been investigated in regard to many different aspects of female reproduction, especially considering that nowadays there is a well-established association between tobacco smoke and female infertility issues. Thus, an experimental study aimed to elucidate the effect of cigarette smoke on reproductive potential by examining the implications of nicotine on follicular growth and vascularization of freely transplanted murine ovarian follicles [40]. Female Syrian golden hamsters were used for all necessary experiments, including the implantation of dorsal skinfold chambers, consequent follicle isolation and transplantation, and daily subcutaneous nicotine treatment of the recipients, assimilating the concentrations observed in more (1.0) or less (0.2 mg/kg) regular smokers. As a consequence, it was brought to attention that nicotine, as a potent toxicant of cigarette smoke, was not able to negatively impact vascularization compared to saline-treated controls, but it did cause inhibition of follicular growth due to a large percentage of granulosa cell apoptosis within the transplanted ovarian follicles of high-dose nicotine-exposed animals, as shown by immunohistochemistry assays for cleaved *Caspase-3*. With follicular growth being a vital parameter of ovulation and fertilization, nicotine-induced apoptosis of granulosa cells could represent one of many mechanisms behind cigarette smoke and fertility-associated diseases, as it is reported in Figure 2.

## 3. The Impact of Tobacco Smoke on the Proliferation of Mammal Granulosa Cells

### 3.1. Induction of Apoptosis in Granulosa Cells

Granulosa cells (GCs) are essential ovarian somatic cells that play an important role in the process of folliculogenesis. At a follicular level, they are located close to the oocyte and their communication is guaranteed via cytoplasmic extensions of corona radiata cells that cross the zona pellucida [41,42]. More specifically, granulosa cells are able to provide nutrients and maturation-inducing factors in order to ensure a successful maturation and developmental competency of the oocytes [43]. They also aim to protect the oocyte from oxidative stress damage [44], considering that this type of cells possesses its own enzymatic antioxidant system, which regulates the levels of reactive oxygen species (ROS) during oocyte maturation [45]. For these reasons, the morphology and number of surrounding granulosa cells have been frequently used as potential biomarkers for developmental competency, as well as embryo and pregnancy outcome [46,47]. In the case of premature rupture of communication between granulosa cells and the oocyte or granulosa cell apoptosis, the developmental competence of the oocyte gets significantly reduced [48,49] and there can be severe degenerative changes [50]. Thus, in order to ensure the acquisition of the developmental competency of follicular COCs [51], the incidence of granulosa cell apoptosis must be kept below the threshold level. Otherwise, a higher incidence of apoptotic granulosa cells could be associated with an increased number of empty follicles, fewer oocyte retrievals, and poor quality of oocytes [52].

Having said that, considering the rise of tobacco use amongst pregnant women, there has been an incrementing interest regarding possible gestational complications or adverse effects directly on the future offspring. Thus, a prospective, randomized study, aiming to investigate the intrauterine impact of cigarette smoke on cell death and DNA damage in murine granulosa cells, was conducted [53]. Twenty-five female Wistar-albino rats were divided into two groups (*n* = 13, *n* = 12), exposed, correspondingly, to cigarette smoke or room air for specific time frames. The recipients of tobacco smoke were exposed to a total of 10 cigarettes daily, 1 h twice a day, for a duration covering their proestrus period all the way up to their effective pregnancies. The newborn offspring were later categorized as Group 1 and Group 2, equivalent to the ones exposed to tobacco smoke during their intrauterine life and the ones that had only had contact with room air while still in utero. In Group 1, a more prominent overall apoptotic morphology with pyknotic, condensed nuclei inside the cells was immediately highlighted by haematoxylin stain. The immunohistochemical slide evaluation confirmed the presence of cytoplasmic and nuclear Caspase-3 immunostaining in granulosa cells of rats that had undergone tobacco smoke exposure. In addition, the immunofluorescent TUNEL-based technique also demonstrated increased DNA damage in granulosa cells, as well as the ovarian surface epithelium in Group 1. In combination with a final quantitative analysis via HSCORES calculation, these findings clearly sustained that there had been an important increase in the incidence of granulosa cell apoptosis, detected by *Caspase-3* immunostaining, in the first group, following a prolonged period of contact with tobacco-generated smoke. As a consequence, it could be stated that cigarette smoke-associated intrauterine exposure can be responsible of significantly decreasing the ovarian reserve of the female progeny, raising fundamental concerns and the need for future investigations regarding the transgenerational effects of maternal smoking on the ovarian function in humans.

There is still a limited amount of knowledge when it comes to the underlying mechanisms of tobacco smoke-induced infertility. For this reason, an experimental study sought to better comprehend the causes behind smoking and ovotoxicity in terms of follicular reduction, apoptosis, and oxidative stress by using a direct nasal exposure mouse model of cigarette smoke-induced chronic obstructive pulmonary disease [54]. Twenty-seven 5-week-old C57BL/6 mice were nose-only exposed to cigarette smoke by 12 cigarettes twice a day, five times per week for a total of 12 to 18 weeks. Each exposure cycle lasted 60 min. Mice that served as controls were exclusively kept in contact with room air. It was observed that following exposure to tobacco smoke, there were increased levels of primordial follicle depletion and antral follicular apoptosis, with markers of early follicular atresia, such as *p53* and *Casp3*, being localized in the granulosa cells of the smoke-exposed antral follicles. In addition to these apoptotic events, there were also evident signs of persistent oxidative stress both in the exposed ovaries and ovulated oocytes that escaped destruction, with the CYP2E1 detoxifying enzyme resulting significantly up-regulated, as well as increased levels of mitochondrial ROS and lipid peroxidation, causing a reduced fertilization potential and overall oocyte dysfunction. Ovarian tissue microarray analysis correlated these insults to a sophisticated ovotoxicity mechanism involving various genes, associated, for example, with immune cell-mediated apoptosis, inflammation, or also follicular activation.

### 3.2. Activation of Autophagy-Mediated Pathways and Granulosa Cell Death

Autophagy represents a crucial intracellular mechanism and is evolutionarily conserved from yeast to mammals. The origin of the word derives from the Greek meaning “eating of self” and it was first systematically characterized by Christian de Duve during his seminal work on the discovery of lysosomes over 50 years ago. The process consists of the removal and recycling of damaging organelles and proteins on behalf of the cell itself, in order to ultimately be protected from destruction. More specifically, a portion of the cytoplasm gets enveloped in double membrane-bound structures called autophagosomes, which undergo maturation and are consecutively fused with lysosomes for degradation [55,56]. Consequently, these events are able to ensure the survival of the cell as an adaptive response to stressful stimuli and can be indirectly considered a form of protective mechanism. Autophagy can be potentially mediated either by nutrient depletion or energy exhaustion [57] and is particularly known to be implicated in a series of diseases like cancer [58], where it is required for tumor cells to survive metabolic stress [59], as well as inflammatory and neurodegenerative disorders. To date, multiple autophagic regulators have been identified, such as Beclin 1 (*BECN1*), microtubule-associated protein 1 light chain 3 (*LC3*), and B-cell lymphoma 2 or *BCL2*. More importantly, however, the results of many studies have illustrated that, in addition to being a temporary stress-associated defensive mechanism, autophagy also promotes cell death by excessive self-digestion and degeneration of essential cellular components [60]. Considering that granulosa cells represent the primary site of apoptosis during follicular atresia, it could be sustained that autophagy may be involved in the process of ovarian folliculogenesis [61]. Nevertheless, the importance of autophagy in terms of toxicant-induced alterations in ovarian function is still substantially unexplored.

Cigarette smoke contains thousands of toxic chemical substances, able to severely affect female reproduction by causing subfertility, premature ovarian failure, and decrease of the primordial follicular pool. However, the mechanism of action behind these events is still largely unknown. Hence, a study was specifically designed to elucidate the underlying follicular loss-correlated mechanisms following exposure to tobacco smoke [13]. The effects were studied in 8-week-old female C57BL/6 mice, which were exposed to cigarette-generated smoke twice a day, 5 days a week for a total of 4, 8, 9, or 17 weeks, using a whole-body smoke exposure system. Exposure for all four time periods resulted in significant reductions in the follicle number of different developmental stages compared to sham controls. Interestingly, at 8 weeks, an increased number of autophagosomes were detected in the tobacco-smoke exposed murine ovarian granulosa cells, alongside a higher expression of key regulatory proteins involved in the Atg autophagic pathway, such as Beclin-1 and microtubule-associated protein light chain 3. Still, following immunohistochemical staining of both sham and pre-treated ovaries for *Bcl-2* and *Bax*, despite a noticeable decrease in the expression of the anti-apoptotic *Bcl-2* in the exposed ovaries, there was no increase in the expression of proapoptotic mediator *Bax*, resulting in a failure to induce apoptosis. Expression levels of the small heat shock protein HSP25 were increased, while at the same time expression of the superoxide dismutase 2 protein was found to be decreased in tobacco-smoke exposed mice, indicating a lower ability to deal with reactive oxygen species or ROS. In combination with each other, these results clearly indicated that, in the given experimental conditions, cigarette smoke exposure did not induce a *Bcl-2*/*Bax*-related granulosa cell apoptosis, but it rather managed to activate the Atg autophagic pathway, ultimately leading to murine ovarian follicle loss.

Taking it a step further, the same research group subsequently tried to test whether tobacco smoke is also able to provoke a dysregulation of the mitochondrial repair mechanisms, causing loss of ovarian follicles via autophagy-mediated granulosa cell death [62]. Eight-week-old female C57BL/6 mice were exposed to cigarette smoke twice a day, 5 days a week for a total of 8 weeks via a whole-body exposure system. Murine ovaries were collected from both sham and smoke-exposed animals and initially processed for transmission-electron microscopy, also known as TEM. It was seen once again that autophagosomes were found to be much more abundant in granulosa cells of animals that had been in contact with tobacco smoke compared to their corresponding control group. Moreover, the autophagy cascade proteins, BECN1 and LC3, resulted in being upregulated, whereas their antagonist and known autophagy inhibitor, BCL2, was downregulated following exposure of the murine granulosa cells, indicating the induction of the autophagic process. In terms of mitochondrial distress, quantitative real-time PCR was performed for three specific mitochondrial repair mechanism markers, *Parkin*, *Mfn1,* and *Mfn2*. Such analysis showed a prominent increased expression of parkin, as well as a clear decreased expression of the mitofusins, suggesting that exposure to cigarette smoke is able to promote actual mitochondrial damage. Considering these results, it could be sustained that exposure to tobacco smoke actively induces a repair dysfunction of the mitochondria, leading to activation of the autophagic machinery and consequent follicle death.

Nonetheless, further attempts have been made to specifically detect the molecular targets of smoke-induced activation of the ovarian reparative autophagy pathway [63]. According to another experimental study, female C57BL/6 mice were kept in direct contact with cigarette smoke twice a day, 5 days a week for an 8-week-time period thanks to a mainstream smoke exposure system. Ovaries were collected for histology and follicle count and their remaining parts were promptly frozen and stored at −80 °C for gene and protein expression. An autophagy gene array indicated that, as a result of smoke exposure, there had been a higher than 2-fold increase in the expression of a series of proautophagic genes, such as *Cdkn1b*, *Map1lc3a*, *Bad*, and *Sqstm1/p62* compared to controls never exposed to cigarette smoke. Quantitative real-time PCR also managed to detect a significant increase in *Prkaa2*, *Pik3c3*, and *Maplc31b* expression, as well as a noticeable decrease in the *Akt1* and *Mtor* expression in smoke-exposed animals compared to their corresponding control group. In terms of western blot-measured protein expression from whole-ovarian homogenates for either sham- or smoke-exposed mice, the 5′-AMP-activated protein kinase catalytic subunit (AMPK) alpha1 + alpha2 and ATG7 expression levels were substantially increased, while the ones of AKT1, mTOR, CDKN1B/p27, and CXCR4 proteins resulted being decreased in cigarette smoke-exposed versus control ovaries. The up-regulation of both AMPK alpha1 + alpha2, a well-established initiator of autophagic signaling, and ATG7 can only furtherly underline the activation of the autophagic cascade. In addition, AKT and mTOR, two known prosurvival factors, were found to be severely decreased in expression, providing another outcome capable of favoring the induction of the autophagic machinery. In summary, this kind of data can verify that cigarette smoke exposure is, indeed, able to cause ovarian follicular loss, as well as mitochondrial dysfunction and oxidative stress, leading to an inability on behalf of the murine granulosa cells to meet their energy needs. As a consequence, this culminates in the activation of the reparative autophagic cascade via the AMPK pathway, alongside inhibition of the anti-autophagic markers, AKT and mTOR.

Considering the highly hazardous consequences of these events on female fertility, an experimental study aimed to elucidate if a specific medication, named chloroquine (CQ) or its derivative hydroxychloroquine, was able to potentially inhibit the cigarette smoke-induced activation of autophagy in the mouse ovary [64]. For this purpose, eight-week-old C57BL/6 mice were implanted with CQ pellets, containing 0.25 and 50 mg CQ/kg. Half of the animals in each group were exposed to cigarette smoke twice a day for 8 weeks, while the other half solely to room air. At the end of the exposure period, ovaries were harvested for electron microscopy, as well as gene and protein expression analysis. Once more, it was immediately observed that autophagosomes were more abundant in granulosa cells of ovaries from the tobacco smoke-exposed mice compared to the room-air exposed controls. However, the CQ treatment at both pellet-contained concentrations caused a significant decrease of the granulosa cell smoke-induced autophagosomes and attenuated the effects of pro-autophagic LC3B and BECN1 expression. These results could be indicative of a possible use of chloroquine in order to attenuate smoke-related autophagic events in the ovary, as well as of the fact that ovarian protection against toxic insult may be eventually achievable.

## 4. Electronic Cigarettes and Female Reproduction

For a complete elaboration of the topic, some data related to the effect of electronic cigarettes on the female reproductive system are henceforth reported. Nonetheless, a review of the literature does not highlight the existence of data associated specifically to mammal granulosa cells, which represent the main focus of the current review. Still, the use of e-cigarettes is steadily considered of rising importance from a reproductive point of view, especially considering that they have recently become available in the market as effective smoking cessation tools [65,66,67], or they often appear to represent a seemingly safe alternative to traditional tobacco use. Electronic cigarettes, also known as e-cigarettes or e-cig for short, are devices capable of vaporizing a nicotine solution instead of burning tobacco leaves. In addition to nicotine, the composition of the e-cig-generated aerosol consists of a mixture containing glycols, metals, aldehydes, polycyclic aromatic hydrocarbons, and volatile organic compounds [68], which are all inhaled during the use of this specific device. Despite well-established evidence that smoking during pregnancy can be responsible for a variety of obstetrical and neonatal complications, such as preterm labor and low birth weight of the offspring [69,70], data reports that 10–35% of women on a global scale continue to smoke during gestation [71,72,73,74] due to potential complete abstinence challenges that they may face. The global prevalence of smoking among pregnant women has been estimated at 1.7% and on a regional level, highest in Europe at 8.1% and lowest in Africa at 0.8%. The three counties with the highest prevalence of smoking during pregnancy are Ireland (38%), Uruguay (29%), and Bulgaria (29%) (Figure 3) [75]. In a recent study carried out in pregnant smokers, it was also seen that the use of e-cigarettes as a solution to quit tobacco smoking was much more common than any other FDA-approved smoking cessation tool [76]. For this reason, both fertility and embryo implantation trials have been performed in mice following exposure of the animals to e-cigarette-produced aerosol for 4 months [77]. Exposed dams showed a significant delay in the onset of the first litter and a noticeable impaired embryo implantation despite the detection of high progesterone levels, a known pregnancy indicator. In addition, female offspring exposed to e-cigarette smoke in utero exhibited an important weight reduction at 8.5 months, while, on the other hand, in utero-exposed adult males demonstrated a slight but non-significant fertility and pup number reduction. Another study furtherly aimed to evaluate the impact of this type of smoking on placental trophoblast function via exposure of HTR-8/SVneo cells to unflavored e-cigarette vapor-conditioned media [78]. While there was no detection of deriving effects on cell viability, proliferation or migration, e-cigarette conditioned media caused a significant reduction in trophoblast invasion and tube formation. As a consequence, all the above could indicate that e-cigarettes are capable of impairing pregnancy initiation and fetal health, further suggesting that their use on behalf of reproductive-aged women or during pregnancy should be considered with extreme caution. Taking into consideration the vital importance of granulosa cells to folliculogenesis and oocyte maturation, as well as pregnancy and embryo development, exposure of these cells to e-cigarette-generated smoke should also be urgently investigated in future studies.

## 5. Conclusions

Substantial harmful effects of cigarette smoke on fecundity and reproduction have become apparent. There is much evidence of a causal relationship with spontaneous abortion, preterm delivery, fetal growth restriction, placenta previa, placental abruption, ectopic pregnancy, sudden infant death syndrome, and congenital anomalies [3]. In this review of the literature, we reported several data of serious biological and clinical implications that can originate from exposure of female reproductive cells to cigarette smoke, based on the description of how smoking and tobacco constituents are able to specifically exert noxious effects on the activity of mammal granulosa cells. Granulosa cells are the functional unit of the ovary, playing a pivotal role in the regulation of ovarian reproductive health by a bidirectional cross-talk with the oocyte. It has been demonstrated that tobacco constituents are able to cause alterations in luteal steroidogenesis, possibly affecting progesterone and estradiol production, as well as suppression of the OCC expansion. In addition, a follicular number reduction and ovotoxicity consisting mainly of primordial follicle activation, follicular atresia, and mitochondrial damage have been found. An increase in the incidence of apoptosis and activation of autophagy has also been manifested after cigarette smoke exposure. The cause–effect correlation of all the above can be seen in Figure 4. In spite of everything, the effects of cigarette smoke on fecundity and newborn outcomes are not generally appreciated. According to reports, 10 to 35% of women worldwide continue to smoke during pregnancy [71,72,73,74] even if reducing smoking in pregnancy is a policy priority in many countries. Research indicates that complete smoking cessation in early pregnancy is most effective in decreasing the risk of adverse newborn outcomes, including preterm birth and low birth weight [79]. A number of studies have demonstrated a dose-dependent adverse effect of smoking on fertility and a possible reversible nature of some effects, providing an important educational and motivational tool that may help to convince current smokers to stop. Women are becoming increasingly aware that smoking during pregnancy is unsafe and acknowledge the potential harms of smoking, but they may not fully recognize the seriousness of smoking effects on the health of a child and believe that it is acceptable to smoke during pregnancy [79]. Therefore, cessation interventions implemented before conception are best. A big portion of women uses electronic cigarettes as an alternative smoking cessation tool. Although reproductive trials have demonstrated that also e-cigarettes can be responsible for embryo implantation and development impairments in mice, studies have not yet been performed specifically regarding e-cig-associated exposure in mammal granulosa cells. Considering all previous findings and their vital importance to oocyte quality and future pregnancy, this could indicate a need to carry out future scientific investigations in this type of cells. For a complete discussion of the matter, it has been considered that several environmental and lifestyle factors (stress, physical activity, alcohol intake, shift work) are known to negatively impact female fertility, and, in many cases, they seem to influence the occurrence of epigenetic modifications with implications with transgenerational effects [80]. Due to maternal smoking, for instance, it has been seen that DNA methylation at specific CpG sites of sorted cord blood CD4+ cells of their newborns can be altered, with hundreds of differentially methylated regions resulting overrepresented in vital regulatory units. In addition, cigarette smoke could also cause miRNA modifications in the human placenta, possibly altering embryonic gene expression [81]. This issue has not been discussed here because at our knowledge there are no available studies in existing literature reporting smoking exposure-induced epigenetic modifications in granulosa cells. In conclusion, it is imperative that the relationship between lifestyle factors and fertility continues to be explored as to reduce the morbidity connected with infertility.

## Figures and Tables

**Figure 1 biomedicines-08-00309-f001:**
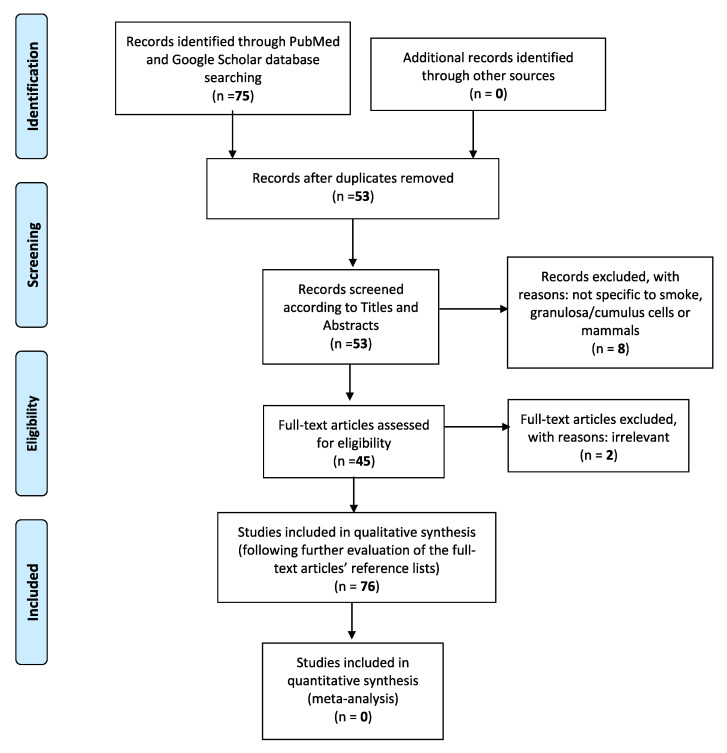
The Preferred Reporting Items for Systematic reviews and Meta-Analyses (PRISMA) flow diagram identifies the total number of articles initially surveyed and the number of articles included and excluded for this systematic review.

**Figure 2 biomedicines-08-00309-f002:**
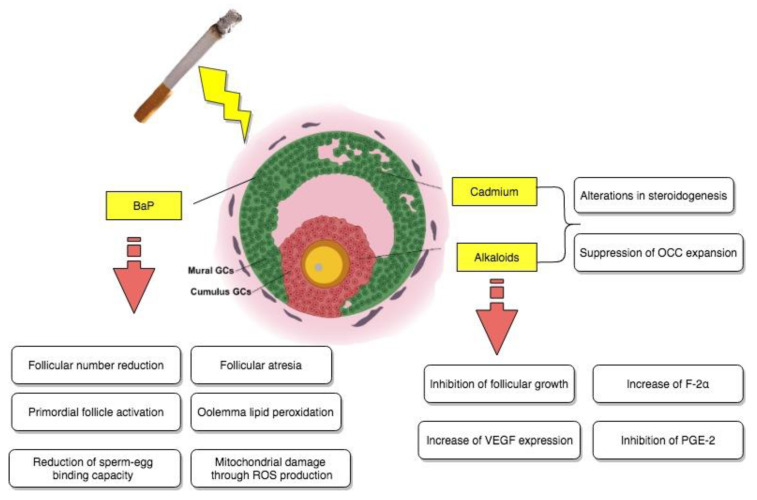
Direct impact of tobacco smoke-contained chemical substances, cadmium, BaP, and alkaloids on granulosa cells.

**Figure 3 biomedicines-08-00309-f003:**
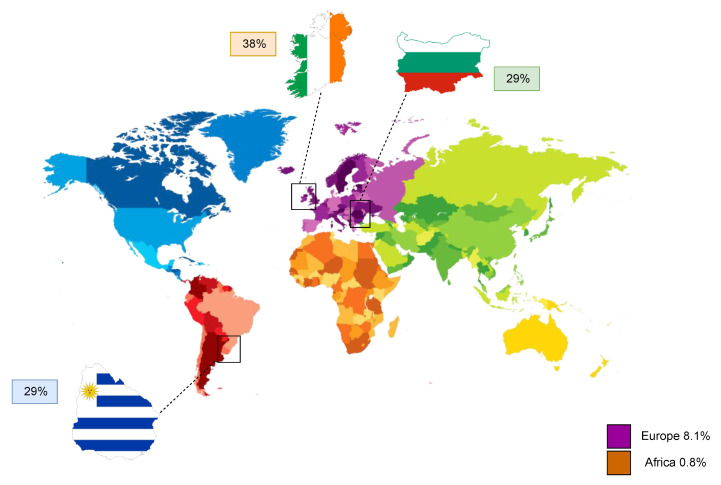
Regional prevalence of cigarette smoking during pregnancy in Europe and Africa, as well as the top three countries, Ireland, Uruguay, and Bulgaria, with the highest known incidence worldwide.

**Figure 4 biomedicines-08-00309-f004:**
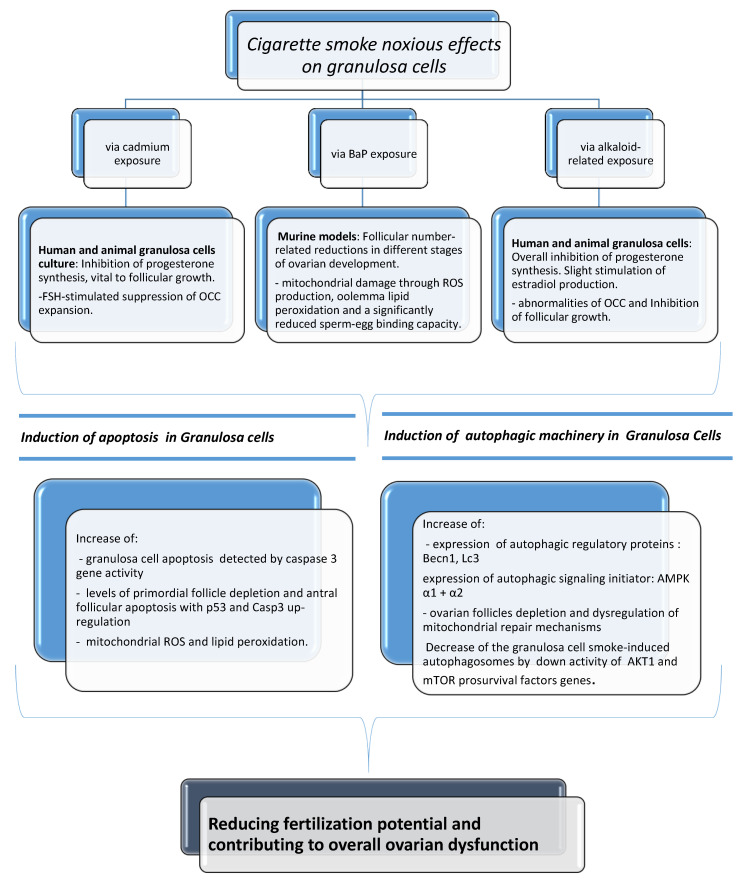
The noxious effects of cigarette smoke in mammal granulosa cells via cadmium, BaP, and alkaloid-related exposure and relative molecular pathways involved.

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
