# Peer review of "Looking Inside the World of Granulosa Cells: The Noxious Effects of Cigarette Smoke"

_biomedicines, 2020, doi:10.3390/biomedicines8090309_

Round 1

Reviewer 1 Report

In this manuscript, the authors reported their review of the implications of cigarette smoking on granulosa cells. This review has merit, however, it requires many improvements. The following concerns should be taken into consideration in their revised manuscript.

Section 1  Give the reference for the effects of Cigarette smoking on wildlife.

The authors have mentioned that cigarette contains Cd, alkaloid, etc.. There are also other heavy metals such as Lead, also arsenic, benzene, Uranium, etc..

Did the authors' used search terms combined with the option “OR” to maximize the yield of relevant articles

Why did the authors restrict the search to two search bases?

Figure 1 title is the search strategy.

Emphasis the inclusion and exclusion criteria clearly

How did the authors select the studies? What was the percentage of titles in agreement with the search keywords?

Cite the figure 2 in the text.

There are studies conducted on cigarette induced epigenetic alterations (Zong et al., 2019. BMC) and there is much more literature.

Reviewer 2 Report

The manuscript of Konstantinidou and colleagues is an original and innovative overview of the effects of some derivatives of cigarette smoke on granulosa cells.

In general, the review is linear, concise, clear, simple and offers potential reflections on the effects of electronic cigarettes.

My only suggestions are:
1) enrich the manuscript with other figures to make it more attractive

2) There are some typos in the text to correct

3) Regarding reference 27, I suggest the authors to see the manuscript: Kumar A et al. Cannabimimetic plants: are they new cannabinoidergic modulators ?. Planta. 2019
